# Basic neonatal resuscitation skills of midwives and nurses in Eastern Ethiopia are not well retained: An observational study

**Yitagesu Sintayehu** ⓘ *, **Assefa Desalew** ⓘ, **Biftu Geda, Getahun Tiruye, Haymanot Mezmur, Kasiye Shiferaw, Teshale Mulatu**

School of Nursing and Midwifery, College of Health and Medical Sciences, Haramaya University, Harar, Ethiopia

* yitagesu.sintayehu@gmail.com

## Abstract

### Background

Neonatal resuscitation is a life-saving intervention for birth asphyxia, a leading cause of neonatal mortality. Worldwide, four million neonate deaths happen annually, and birth asphyxia accounts for one million deaths. Improving providers' neonatal resuscitation skills is critical for delivering quality care and for morbidity and mortality reduction. However, retention of these skills has been challenging in developing countries, including Ethiopia. Hence, this study aimed to assess neonatal resuscitation skills retention and associated factors among midwives and nurses in Eastern Ethiopia.

### Methods

An institution-based cross-sectional study was conducted using a pre-tested, structured, observational checklist. A total of 427 midwives and nurses were included from 28 public health facilities by cluster sampling and simple random sampling methods. Data were collected on facility type, availability of essential resuscitation equipment, socio-demographic characteristics of participants, current working unit, years of professional experience, whether a nurse or midwife received refresher training, and skills and knowledge related to neonatal resuscitation. Binary logistic regression was used to analyse the association between neonatal resuscitation skill retention and independent variables.

### Results

About 11.2% of nurses and midwives were found to have retention of neonatal resuscitation skills. Being a midwife (AOR, 7.39 [95% CI: 2.25, 24.24]), ever performing neonatal resuscitation (AOR, 3.33 [95% CI: 1.09, 10.15]), bachelor sciences degree or above (AOR, 4.21 [95% CI: 1.60, 11.00]), and good knowledge of neonatal resuscitation (AOR, 3.31 [95% CI: 1.41, 7.73]) were significantly associated with skill retention of midwives and nurses.

**Data Availability Statement:** All related data are presented fully within the paper and its Supporting Information files.

**Funding:** CIRHT-Ethiopia provided support in the form of per diem for data collectors, supervisors and provided training for the team. No additional external funding was received for this study.

**Competing interests:** The authors have declared that no competing interests exist.

**Abbreviations:** NR, Neonatal Resuscitation; WHO, World Health Organization 31769.

## Conclusion

Basic neonatal resuscitation skills of midwives and nurses in Eastern Ethiopia are not well retained. This could increase the death of neonates due to asphyxia. Being a midwife, Bachelor Sciences degree or above educational status, ever performing neonatal resuscitation, and good knowledge were associated with skill retention. Providers should be encouraged to upgrade their educational level to build their skill retention and expose themselves to NR. Further, understanding factors affecting how midwives and nurses gain and retain skills using high-level methodology are essential.

## Introduction

Worldwide, 136 million babies are born annually. Ten million require some stimulation at birth to breathe, while six million require basic resuscitation with a bag and mask [1,2]. Evidence shows that one million neonatal deaths occur on the day of birth per year. Close to two million babies die in the first week of life and four million dies in the neonatal period [3–5], which accounts for 46% of under-five mortality [6–8]. This mortality is estimated to increase to 52% in 2030 [9,10], unless strategic interventions are implemented. Surprisingly, about 99% of neonatal deaths occur in low-income and middle-income countries [1]. Unfortunately, four million neonates experience birth asphyxia and one million deaths are caused by birth asphyxia annually [11–18], while resuscitation at birth can prevent a large proportion of mortality [11,19].

Neonatal resuscitation (NR) is a crucial intervention for birth asphyxia [20]. In low- income and middle-income countries, facility-based NR may avert 30% of intrapartum-related neonatal deaths [3,21]. NR training for health workers potentially prevents many neonatal deaths currently misclassified as still births [22]. NR with a bag and mask is a high-impact intervention that can reduce neonatal deaths in resource-limited countries [23].

The burden of neonatal death is particularly severe in sub-Saharan Africa, including Ethiopia, which suffers from a current neonatal mortality rate of 29 deaths per 1000 live births annually [1,2,24,25]. To provide effective resuscitation, care providers should apply detailed information and employ their technical skills competently. Therefore, those who perform resuscitation, nurses and midwives are expected to be competent and have the required knowledge and skills. These skills should be learned through training during their academic courses. Despite the time, resources, and expenses allocated to such training, little is known about the outcomes of these educational programs and whether students' educational needs are met.

Simulation-based assessment of NR skills to identify providers' strengths and weaknesses is a less costly and more practical alternative to direct observation of actual NR [23]. The delivery of effective, safe, and good quality health services is essential interventions to improving maternal, newborn, and child health. Skill retention has paramount importance to improving quality service delivery [26].

Moreover, to achieve sustainable development goals (SDG) in 2030, all countries should reduce neonatal mortality to 12 per 1,000 live births and under 5 mortality to at least as low as 25 per 1,000 live births [27]. Retention of NR skills is of crucial importance, along with access to basic equipment to save the life of millions of newborns [23,28,29]. However, maintaining NR skills presents a different set of challenges, particularly in settings where providers attend few deliveries, infrequently resuscitate newborns, and have limited access to refresher training in resource-scarce settings, including Ethiopia. Available evidence indicates that the rapid loss of NR skills occurs after an initial training [21,28,30–33].

Midwives and nurses are healthcare providers who are able to provide promotive, preventive, curative, and rehabilitative services at all levels of the health care system. Midwifery and nursing training in Ethiopia offered in both public and private per the ministry of science and higher education requirement: the diploma, post-basic, direct Bachelor of Sciences (BSc) degree, MSc degree, and PhD level in both qualifications. Since nursing professionals are assigned to all units of the health facility, their education focuses on all medical care. Even if midwives take all medical care education, lastly they focus on maternal and neonatal care. In addition to these, some midwives and nurses take in-service training that can help them to give quality care at where they are assigned. Therefore, both midwives and nurses have access to managing critically ill newborns, midwives in the labor and delivery ward and nurses in delivery, neonatology, and pediatric ward.

NR skills are a key component of efforts to reduce neonatal morbidity and mortality. Although appropriate neonatal care programs are a current issue in Ethiopia, evidence on NR skill retention among midwives and nurses is scarce. Therefore, this study aimed to assess the magnitude of NR skills retention and associated factors among midwives and nurses in Eastern Ethiopia.

## Materials and methods

### Study setting and period

The study was conducted in Eastern Ethiopia, specifically in health care facilities from the Harari regional state, the Dire Dawa administration, and the Eastern and Western Harerghe zones from the Oromia regional states. In the included locations, there were an estimated 6,565,406 total population and 14 hospitals and 221 health centers with a total of 521 midwives and 2735 nurses working in all public health facilities in Eastern Ethiopia in November 2018.

### Study design and participants

A facility-based cross-sectional study was conducted among midwives and nurses who were working in public health facilities and were employed for at least one year anywhere in the study area were included. A total of 437 midwives and nurses were expected to be included in the assessment. Of these, 10 midwives and nurses were not included because of different reasons (four participants refused the invitation and the remaining six participants were inaccessible because they were in annual leave at the time of the assessment. Therefore, the assessment was limited to 427 (97.7%) of midwives and nurses those completely addressed by the assessment team. The assessment team visited each facility for one to three days to collect data if the study participants were unavailable.

### Sample size determination and sampling procedures

The sample size was determined using a single population proportion formula ($n = \frac{(Z_{\alpha/2})^2 p(1-p)}{d^2}$ by considering the proportion of NR skill retention of healthcare providers (76.3%) [34], 95% confidence level z values of 1.96, a margin of error of 5%, and adding a 5% allowance for non-response. Since the study had two stages, we used a design effect of 1.5. The final sample size resulted in 437 study participants. We used a cluster-sampling method first, based on their geographic location and administration into the Dire Dawa Administration (desert/hot climate and urban), Harari regional state (urban and middle climate region), and East Hararghe and West Hararghe (rural with different geographic locations and administrations). Then, we selected a total of 10 hospitals and 18 health centers from those clusters using a simple random sampling method from the list of public health facilities. In addition, a simple random sampling (lottery) method and proportional allocation to size were used to enroll midwives and nurses from the selected facilities.

## Data collection techniques

A structured questionnaire was adapted and modified from similar literatures. The knowledge questionnaire was based on model assessment forms in the Needs Assessment Toolkit (the Averting Maternal Death and Disability Program's Needs Assessment toolkit) regarding steps providers should take during newborn resuscitation [23, 35–38]. Each knowledge question was designed to elicit multiple responses (25 expected responses). The tool was pre-tested and modified to suit the local context. The data collection procedure was undertaken using the pre-tested face-to-face interview administered questionnaire using paper-printed English version language. First, we interviewed facility directors and the study subjects about facility characteristics like the type of facility (hospital or health center), availability of essential NR equipment, current participants working unit, years of professional experience, refresher training status, and knowledge questions. Then, experts observed participants performing NR on an anatomical model and used an observational checklist (38-item skills tools) to rate their skill level [23, 39, 40].

## Measurement of knowledge and skills

Knowledge of midwives and nurses on neonatal resuscitation was determined using a set of 25 objective questions. Each correct answer was valued at one point, and incorrect answer scored zero points. Questions not answered were considered incorrect answers. Eventually, participants were then grouped into two categories based on their total score on the knowledge scale: good knowledge (score 80% or higher) and poor knowledge (score less than 80%), which have been used in different studies after the overall cumulative mean score and the cumulative mean percentage score were calculated [37, 41–44].

Similar to knowledge measurement, the skill of midwives and nurses on neonatal resuscitation was determined using a set of 38 steps of the objective observational checklist. Each complete performed step was valued at one point, and an incomplete performed step/not performed steps was valued zero point. Eventually, participants were then grouped into two categories based on their total score on the skill scale, good retention of skill (score 80% or higher), and poor retention of skill (score less than 80%), after the overall cumulative mean score and the cumulative mean percentage score were calculated [41].

## Data quality control and management

Three experts (midwifery department lecturers, one Master in neonatology and pediatrics, and two Master in maternity and neonatology) were trained for observation. Three interviewers (BSc degree in midwifery) and two Master of public health (MPH) supervisors received intensive training for three days about the purpose of the study, the sampling procedure, and observational skills. All were experienced service providers and had data collection experience in a maternal and neonatal related study. Before data collection, the principal investigators shared ethical issues and ways of addressing contingency management. The questionnaire was pre-tested on 5% (22 participants) of the sample size on a similar population other than the included facility (in one hospital and two health centers). Then, all completed questionnaires were checked for completeness and cleaned manually.

## Data processing and analysis

The questionnaires were coded and entered into the EPI data version 3.1 statistical software and then exported to SPSS version 21 statistical software for analysis. Data were summarized and presented using descriptive statistics. The outcome variables were coded as "1" for having

good retention whereas "0" for not having good retention. The association between the outcome variables (i.e., NR skill retention) and independent variables were analyzed using a binary logistic regression model. Covariates with a p-value<0.25 were retained and entered into the multivariable logistic regression analysis using a forward stepwise approach. The Hosmer and Lemeshow goodness-of-fit test was used to assess whether the necessary assumptions were fulfilled. The results are presented as odds ratios OR with 95% confidence intervals. An association with a p-value<0.05 was considered significant.

## Ethical considerations

Ethical clearance was obtained from the Institutional Health Research Ethics Review Committee of Haramaya University in Ethiopia. Following the approval, an official letter of cooperation was given to the Region and City administration Health Bureaus, zonal and Woreda health offices, hospitals, and health center officials. Permission was obtained from each facility's director prior to the study. The purpose and importance of the study were explained to the study participants and directors of the facilities. Overall, data were collected only after fully written informed consent was obtained from each participant. All findings were kept confidential. The names and addresses of the participants were not recorded in the questionnaire. Furthermore, all the basic principles of human research ethics (respect for persons, beneficence, voluntary participation, confidentiality, and justice) were respected.

## Results

### Sociodemographic characteristics of participants

A total of 427 study subjects participated, yielding a response rate of 97.7%. The mean (SD) age of participants was 28.4 (± 7.05). Two hundred fifty-eight (60.4%) were female and 245 (57.4%) were married. Two hundred (46.8%) were Orthodox Christian, 285 (66.7%) were nurses, and 292 (68.4%) had a bachelor sciences degree or above. Two hundred sixty-three (61.6) of them had 1–5 years of professional experience (Table 1).

### Training and work-related exposure

One hundred seven (25.1%) participants had ever taken NR refresher training and 202 (47.3%) had ever performed NR. About half (50.1%) of study subjects ever worked in the delivery room (Fig 1).

### Health facilities characteristics

According to the study, 127 (29.7%), 107 (25.1%), 99 (23.2%) and 94 (22%) study participants were from Dire Dawa city administration, Harari regional state, and Eastern and Western Hararghe zone respectively. Furthermore, 97.4%, 63% and 85% of health facilities had neonatal resuscitation corners (NR area), adequate newborn resuscitation guidelines, and essential equipment for newborn resuscitation, respectively. Regarding health care provision facilities, 333 (78%) midwives and nurses were from hospitals and the remaining 94 (22%) were from health centers.

### The magnitude of neonatal resuscitation skill retention

Among participants in this study, only 48 (11.2%) 95% CI (8–14%) had good skill retention.

### Factors associated with the neonatal resuscitation skill retention

In multivariate logistic analyses, being a midwife, having a BSc degree or above, ever performing NR, and having good knowledge of NR were independently associated with skill retention.

**Table 1. Sociodemographic characteristics of study participants among midwives and nurses working in public health facilities in Eastern Ethiopia, November 2018 (n = 427).**

| Variables | Frequency (N) | Percent (%) |
|---|---|---|
| **Age** | | |
| 15–24 | 116 | 27.2 |
| 25–34 | 253 | 59.3 |
| 35–49 | 42 | 9.8 |
| 50–61 | 16 | 3.7 |
| **Sex** | | |
| Female | 258 | 60.4 |
| Male | 169 | 39.6 |
| **Marital status** | | |
| Never married | 171 | 40.0 |
| Married | 245 | 57.4 |
| Divorced/Separated | 11 | 2.6 |
| **Religion** | | |
| Orthodox Christian | 200 | 46.8 |
| Muslim | 179 | 42.0 |
| Protestant | 48 | 11.2 |
| **Profession** | | |
| Midwifery | 142 | 33.3 |
| Nurse | 285 | 66.7 |
| **Professional education levels** | | |
| Level IV diploma | 135 | 31.6 |
| BSc degree and above | 292 | 68.4 |
| **Professional experience in years** | | |
| 1–5 | 263 | 61.6 |
| 6–10 | 114 | 26.7 |
| ≥11 | 50 | 11.7 |
| **Current working unit** | | |
| Obstetrics & gynecology | 290 | 67.9 |
| Pediatrics & neonatal | 137 | 32.1 |

Midwives were seven times more likely to retain NR skills (AOR, 7.39 [95% CI: 2.25, 24.25]) as compared to nurses. Those with a BSc degree and above were four times more likely to retain skills (AOR, 4.21 [95% CI: 1.60, 11.00]). Furthermore, participants who ever performed NR were three times more likely to retain NR skill (AOR, 3.33 [95% CI: 1.09, 10.15]), and those with good knowledge in NR (AOR, 3.31 [95% CI: 1.41, 7.73]) were three times more likely to retain their NR skill when compared with those with poor knowledge (Table 2).

## Discussion

The present study showed that only 11.2% of midwives and nurses had retained the skills of NR. In multivariate logistic regression analysis, being a midwife, having a BSc degree and above educational status, ever performed NR, and having a good level of knowledge were independently associated with NR skill retention of midwives and nurses.

Eleven percent of our sample retained their NR skills. This was lower than studies conducted in India [45, 46], Afghanistan [23], Tanzania [47], and northern Ethiopia [44]. This is probably due to the fact that the other studies were conducted in a single facility and urban facilities, while this study incorporated a broad study setting that included rural facilities that

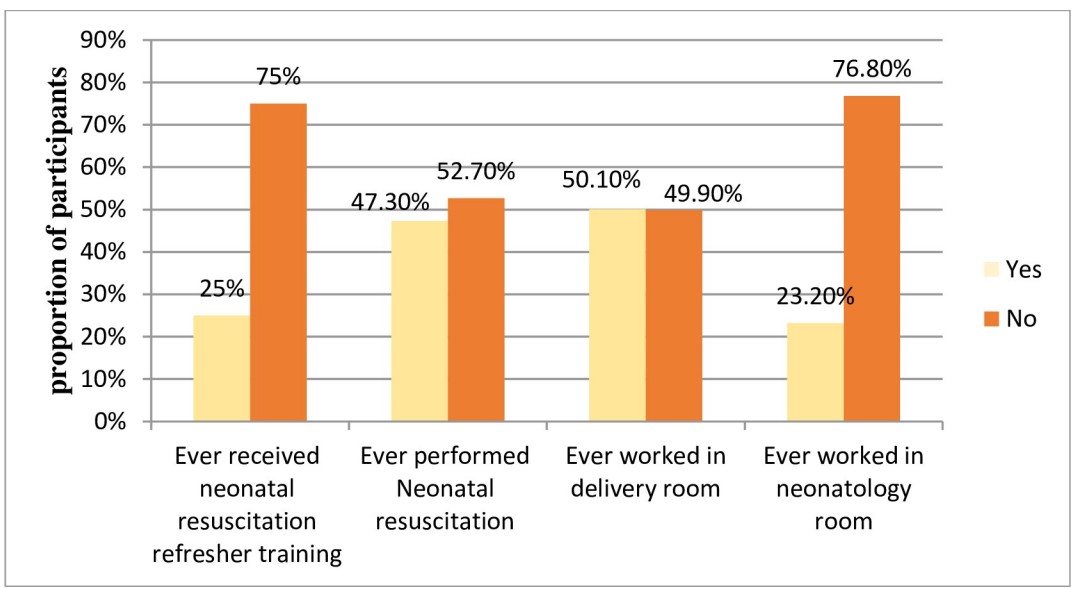

**Fig 1. Refresher training and work-related exposure of midwives and nurses working in public health facilities in Eastern Ethiopia, November 2018 [n = 427].**

had no access to many infrastructures, limited essential equipment, updated NR information and internet access. However, 11.2% reported in our study was greater than some other studies, for example, in Pune city, India (6.3%) [20]. This might be due to the impact of the current focus of neonatal care service in Ethiopia. In Ethiopia, to improve access to neonatal care, many neonatal intensive care unit were opened and midwives and nurses training were started. During this time, a quarter of the midwives and nurses in Eastern Ethiopia received refresher training.

Regarding factors associated with NR skill retention, being a midwife was significantly associated with skill retention; midwives were 7.39 times more likely to retain NR skills as compared with nurses. This might be due to the fact that midwives were frequently exposed in the delivery room and received refresher training. This finding is in agreement with a finding in Afghanistan [23]. In addition, those who ever performed NR were 3.33 times more likely to retain their skill than those not exposed in NR. This finding is in line with studies in Nepal and Italy [48–50].

Furthermore, having higher educational status was associated with better skill retention. Those with a BSc degree and above were 4.21 times more likely to retain their skill than a level IV diploma nurse or midwives. This might be due to the fact that in the curricula of degree learners, there was greater focus on maternal and child health and using evidence-based practice. Moreover, having a good knowledge was found to be significantly associated with the NR skill retention of nurses and midwives. Those with good knowledge about NR were 3.31 times more likely to retain their skill. It is clear that knowledgeable care providers can apply their knowledge to skill. Paradoxically, receiving NR refresher training was not associated with the multivariate logistic analysis of this study, but in many studies, NR training was found to be significantly associated with retention of NR skill [23,30,50–55].

A limitation of this study was the use of anatomic model simulation and observations to assess NR skill. Most providers may not consider and give attention to the anatomical model as a human being. Furthermore, the skill was scored by a single expert, and no inter-rater variation test was conducted. The stress of testing, as well as the stress of clinical situations where

**Table 2. Factors associated with neonatal resuscitation skill retention among midwives and nurses working in public health facilities in Eastern Ethiopia, November 2018 (n = 427).**

| Variables | NR skill retention | | COR | AOR | p-value |
|---|---|---|---|---|---|
| | **Yes** | **No** | | | |
| **Religion** | | | | | |
| Orthodox Christian | 27(13.5) | 173(86.5) | 1 | | |
| Muslim | 18(10.1) | 161(89.9) | 0.72(0.38, 1.35) | 0.64(0.30, 1.34) | 0.23 |
| Protestant | 3(6.2) | 45(93.8) | 0.43(0.12, 1.47) | 0.44(0.11, 1.74) | 0.24 |
| **Profession** | | | | | |
| Nurse | 10(3.5) | 275(96.5) | 1 | | |
| Midwifery | 38(26.8) | 104(73.2) | 10.05(4.82, 20.89) | 7.39(2.25, 24.24) | **0.001**[*] |
| **Level of education** | | | | | |
| Diploma | 6(4.4) | 129(95.6) | 1 | | |
| Degree and above | 42(14.4) | 250(85.6) | 3.61(1.49, 8.72) | 4.21(1.60, 11.00) | **0.003**[*] |
| **Ever received refresher NR training** | | | | | |
| No | 23(7.2) | 297(92.8) | 1 | | |
| Yes | 25(23.4) | 82(76.6) | 3.94(2.12, 7.29) | 2.72(0.81, 3.63) | 0.15 |
| **Ever performed NR** | | | | | |
| No | 6(2.7) | 219(97.3) | 1 | | |
| Yes | 42(20.8) | 160(79.2) | 9.58(3.97, 23.08) | 3.33(1.09, 10.15) | **0.03**[*] |
| **Ever worked in a delivery unit** | | | | | |
| No | 9(4.2) | 204(95.8) | 1 | | |
| Yes | 39(18.2) | 175(81.8) | 5.05(2.38, 10.72) | 0.52(0.15, 1.79) | 0.30 |
| **Ever worked in a pediatrics unit** | | | | | |
| No | 42(12.8) | 286(87.2) | 1 | | |
| Yes | 6(6.0) | 93(94.0) | 0.44(0.09, 0.77) | 0.80(0.22, 2.76) | 0.71 |
| **Level of knowledge about NR** | | | | | |
| Poor knowledge | 31(8.1) | 354(91.9) | 1 | | |
| Good knowledge | 17(40.5) | 25(59.5) | 7.77(3.79, 15.90) | 3.31(1.41, 7.73) | **0.006**[*] |

[*]Statistically significant at p-value <0.05 in multivariate logistic regression analysis.

NR are needed, may have contributed to the low outcome of skill retention. In addition, the study was undertaken at one-time point so it is not possible to establish cause-effect relationships.

## Conclusion

NR skills among midwives and nurses are not well retained in public health care facilities in the Eastern part of Ethiopia, which might increase the death rate of neonates due to asphyxia. Being a midwife, having a BSc degree or above, ever performed NR, and having good knowledge in NR were independently associated with NR skills retention in this study setting. Providers should be encouraged to upgrade their educational level to build their skill retention and expose themselves to NR. Further, understanding factors affecting how midwives and nurses gain and retain skills using high-level methodology are essential.

## Supporting information

**S1 Dataset. The dataset from which the results of the study were produced (SPSS file).** (SAV)

**S1 File. The data collection tool (questionnaire and checklist) in English.**
(DOCX)

## Acknowledgments

The authors would like to thanks CIRHT-Ethiopia for their constructive comments and Haramaya University College of Health and Medical Sciences for their constructive comments and ethical review, and all respective Health Bureau for their cooperation to write permission letters to their health facilities. We also thank the data collectors, participants of the study and Pre-Publication Support Service (PREPSS) supported the development of this manuscript by providing pre-publication peer-review and copy editing.

## Author Contributions

**Conceptualization:** Yitagesu Sintayehu.

**Data curation:** Yitagesu Sintayehu.

**Formal analysis:** Yitagesu Sintayehu, Biftu Geda, Kasiye Shiferaw, Teshale Mulatu.

**Investigation:** Yitagesu Sintayehu, Assefa Desalew, Biftu Geda, Getahun Tiruye, Haymanot Mezmur, Kasiye Shiferaw, Teshale Mulatu.

**Methodology:** Yitagesu Sintayehu, Assefa Desalew, Biftu Geda, Getahun Tiruye, Haymanot Mezmur, Kasiye Shiferaw, Teshale Mulatu.

**Project administration:** Yitagesu Sintayehu.

**Resources:** Teshale Mulatu.

**Software:** Yitagesu Sintayehu, Haymanot Mezmur, Teshale Mulatu.

**Supervision:** Yitagesu Sintayehu, Biftu Geda, Getahun Tiruye, Haymanot Mezmur, Kasiye Shiferaw, Teshale Mulatu.

**Validation:** Yitagesu Sintayehu, Assefa Desalew, Biftu Geda, Getahun Tiruye, Haymanot Mezmur, Kasiye Shiferaw, Teshale Mulatu.

**Visualization:** Yitagesu Sintayehu.

**Writing – original draft:** Yitagesu Sintayehu, Assefa Desalew, Haymanot Mezmur, Kasiye Shiferaw, Teshale Mulatu.

**Writing – review & editing:** Yitagesu Sintayehu, Assefa Desalew, Biftu Geda, Getahun Tiruye, Haymanot Mezmur, Teshale Mulatu.

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
