## [Decision Letter · Decision Letter 0]

7 May 2020

PONE-D-20-09955

Basic neonatal resuscitation skills of midwives and nurses in Eastern Ethiopia are not well retained: An observational study.

PLOS ONE

Dear Yitagesu Sintayehu Abebe,

Thank you for submitting your manuscript to PLOS ONE. After careful consideration, we feel that it has merit but does not fully meet PLOS ONE’s publication criteria as it currently stands. Therefore, we invite you to submit a revised version of the manuscript that addresses the points raised during the review process.

We would appreciate receiving your revised manuscript by June 20 2020. To enhance the reproducibility of your results, we recommend that if applicable you deposit your laboratory protocols in protocols.io, where a protocol can be assigned its own identifier (DOI) such that it can be cited independently in the future. For instructions see: http://journals.plos.org/plosone/s/submission-guidelines#loc-laboratory-protocols

We look forward to receiving your revised manuscript.

Kind regards,

Georg M. Schmölzer

Academic Editor

PLOS ONE

Additional Editor Comments (if provided):

Thank you for your submission. Please find below my comments in addition to those from the reviewer.

Please make sure you upload the research checklist specific for your research methods with your revisions.

Could you please also upload the questionnaire you created, which might allow other researcher to use this in their settings.

Could you please also upload the observational checklist (38-item skills tools)?

Has this observational checklist (38-item skills tools) been validated or used in previous studies?

If available, could you please add the reference number from your ethics application.

I would suggest adding the Socio-demographic questionnaire as an appendix.

thank you

2. We noticed you have some minor occurrence of overlapping text with the following previous publication, which needs to be addressed:

http://jmrh.mums.ac.ir/m/issue_548_669_Volume+3,+Issue+3,+July+2015%3Cspan+id=%22sp_ar_pages%22%3E,+Page+%3Cspan+dir=%22ltr%22%3E378-384%3C/span%3E%3C/article_4464.html?

In your revision ensure you cite all your sources (including your own works), and quote or rephrase any duplicated text outside the methods section. Further consideration is dependent on these concerns being addressed.

3. Please include additional information regarding the survey or questionnaire used in the study and ensure that you have provided sufficient details that others could replicate the analyses. For instance, if you developed a questionnaire as part of this study and it is not under a copyright more restrictive than CC-BY, please include a copy, in both the original language and English, as Supporting Information. Please also include details of any pre-testing that took place, including the number of participants.

"The authors would like to thank CIRHT Ethiopia, for funding the study and technical

collaboration and Haramaya University, College of Health and Medical Science for unreserved

technical and non-financial support, all respective Health Bureau for their support and

collaborations."

Reviewers' comments:

Reviewer's Responses to Questions

**Comments to the Author**

1. Is the manuscript technically sound, and do the data support the conclusions?

Reviewer #1: Partly

2. Has the statistical analysis been performed appropriately and rigorously? 

Reviewer #1: Yes

3. Have the authors made all data underlying the findings in their manuscript fully available?

Reviewer #1: Yes

4. Is the manuscript presented in an intelligible fashion and written in standard English?

Reviewer #1: Yes

5. Review Comments to the Author

Reviewer #1: This study is important due to the high prevalence of neonatal deaths in Africa and the obvious need for adequate neonatal resuscitation skills among local health personnel involved with newborn babies. Some information is lacking, at best too scarce. 1) How was the neonatal resuscitation skills measured, objectively and/or subjectively, by whom and how many evaluated each participant. Interrater reliability? I appreciate the authors mentioning this as a limitation. What was the educational background of the three experts (midwifery department lecturers) that were trained for observation as well as the 3 interviewers/supervisors who received intensive training 2) What is defined as retained skills when a fraction of participants never had participated in training or theoretical background "25 percent had ever taken NR refresher training and 47.3% had ever performed NR. About half (50.1%) of study subjects ever worked in the delivery room". How was skill retention defined and quantified and what is meant by the expression of "good NR knowledge score" 3) What are the characteristics of the 4 health districts, any differences or similarities, in-hospital patients populations? 4) I would like to see the pre-tested, structured, observational checklist for evaluation of NR skills, as well as the adapted and modified structured questionnaire that was used and the scoring system. 5) Further, a better description of how cluster sampling and simple random sampling methods were performed 6) Explain the differences in education and daily exposure to critically ill newborns, midwifes vs nurses, 7) 427 participated in the study. How many were invited and how were the invitation posted. Anyone who rejected participation 8) Minors; Explain NR unit. Fig 2 may be deleted as information appears in the text. Is there an IRB approval number?

6. PLOS authors have the option to publish the peer review history of their article (what does this mean?). If published, this will include your full peer review and any attached files.

Reviewer #1: Yes: Britt Nakstad

---

## [Author Response · Author response to Decision Letter 0]

22 Jun 2020

Response to Reviewers’ 

Title: “Basic neonatal resuscitation skills of midwives and nurses in Eastern Ethiopia are not well retained: an observational study." [Manuscript ID PONE-D-20-09955]

To: The editor-in-chief, PLOS ONE

From: Authors 

Subject: Revision of the manuscript 

Dear Sir/ Madam,

We hope everything is fine. We appreciate and thank to the academic editor and reviewers for investing their time and energy to review and make comments on our manuscript. It is with great pleasure to receive the invaluable and constructive comments for our manuscript. 

As per your request for a separate cover letter, the comments with their point-by-point responses are put below here. In addition, the detailed changes made are highlighted in the “revised manuscript with track changes” by activating the “track changes” feature to easily identify the changes/ improvements. Moreover, the manuscript without track change is prepared. We accepted and tried to incorporate all of the comments provided. Therefore, we are kindly requesting you to review our revised manuscript; especially the “manuscript without track change”. 

No Comments Responses

Editor 

- Additional Editor Comments (if provided):

Thank you for your submission. Please find below my comments in addition to those from the reviewer.

Please make sure you upload the research checklist specific to your research methods with your revisions. Thank you for the comment, the comment is accepted and all sections of the tools used in this study are uploaded as supporting information, this can be possible to upload with link during publication if possible, that might allow other researcher to use this in their settings.

- Could you please also upload the questionnaire you created, which might allow other researcher to use this in their settings. Thank you, the comment is accepted and all sections of the tools used in this study is uploaded as supporting information, this can be possible to upload with link during publication if possible, that might allow other researcher to use this in their settings. 

- Could you please also upload the observational checklist (38-item skills tools)?

Has this observational checklist (38-item skills tools) been validated or used in previous studies? Thank you request. Since we extracted the tools from the different study for both knowledge assessment and skill observation, the detailed explanation and the studies used are listed under the “5. Review Comments to the Author” comment number 4 of reviewer #1.

 If available, could you please add the reference number from your ethics application?

 Our institution did not give IRB approval number since it is not online. The office notices the approval of the study by letter without approval number.

- I would suggest adding the Socio-demographic questionnaire as an appendix.

thank you Thank you, the comment is accepted. All sections of the tools used in this study is uploaded as supporting information, this can be possible to upload with link during publication if possible, that might allow other researcher to use this in their settings.

-

Please ensure that your manuscript meets PLOS ONE's style requirements, including those for file naming. The PLOS ONE style templates can be found at http://www.plosone.org/attachments/PLOSOne_formatting_sample_main_body.pdf and http://www.plosone.org/attachments/PLOSOne_formatting_sample_title_authors_affiliations.pdf

The comment is accepted and corrected accordingly per PLOS ONE’s style.

2. We noticed you have some minor occurrence of overlapping text with the following previous publication, which needs to be addressed: http://jmrh.mums.ac.ir/m/issue_548_669_Volume+3,+Issue+3,+July+2015%3Cspan+id=%22sp_ar_pages%22%3E,+Page+%3Cspan+dir=%22ltr%22%3E378-384%3C/span%3E%3C/article_4464.html?

Thank you for the enquiry. We had gone throughout the linked document; we have tried to paraphrase the overlapped sentence accordingly. Even if the study title is one of the same professionals (midwives and nurses), our study is on in-service providers at health facility level, while their study is on students in skill laboratory. Additionally, their background, result, and discussion section is different in finding plus presentation style. Furthermore, we will try more, if there is any overlapped text are correct.

- In your revision ensure you cite all your sources (including your works) and quote or rephrase any duplicated text outside the methods section. , further consideration is dependent on these concerns being addressed. Thank you. The comment is accepted all sources are cited.

3. Please include additional information regarding the survey or questionnaire used in the study and ensure that you have provided sufficient details that others could replicate the analyses. For instance, if you developed a questionnaire as part of this study and it is not under copyright more restrictive than CC-BY, please include a copy, in both the original language and English, as Supporting Information. Please also include details of any pre-testing that took place, including the number of participants The comment is accepted and a copy of the questionnaire is uploaded as supporting information. Since the participants’ working language is English, we have used only the English language version, and we didn’t have another language version. The questionnaire was pre-tested on 5% (22 participants) of the sample size on a similar population other than the included facility (in one hospital and two health centers). This one is incorporated in the manuscript.

"The authors would like to thank CIRHT Ethiopia, for funding the study and technical

collaboration and Haramaya University, College of Health and Medical Science for unreserved

technical and non-financial support, all respective Health Bureau for their support and

collaborations."

"The funders had no role in study design, data collection, and analysis, decision to publish, or preparation of the manuscript." The comment is accepted and the funding statement removed from the manuscript and corrected as, Acknowledgment:The authors would like to thank CIRHT-Ethiopia for their constructive comments and Haramaya University College of Health and Medical Sciences for their constructive comments and ethical review, and all respective Health Bureau for their cooperation to write permission letters to their health facilities. We also thank the data collectors, participants of the study and Pre-Publication Support Service (PREPSS) supported the development of this manuscript by providing pre-publication peer-review and copy editing.

There is funding statement change as,

Funding Statement: The authors received no specific funding for this work.

 Reviewer #1: Response 

- This study is important due to the high prevalence of neonatal deaths in Africa and the obvious need for adequate neonatal resuscitation skills among local health personnel involved with newborn babies. Some information is lacking, at best too scarce. Thank you!

1) How was the neonatal resuscitation skills measured, objectively, and/or subjectively, by whom and how many evaluated each participant. Interrater reliability? I appreciate the authors mentioning this as a limitation. What was the educational background of the three experts (midwifery department lecturers) that were trained for observation as well as the 3 interviewers/supervisors who received intensive training Thank you, all the raised comments, measurements of both the skill and knowledge are objective assessment tool, Interrater reliability was raised as a limitation (single observer used), and the educational background of data collectors, observers (experts) included in the manuscript. All were experienced service providers and had data collection experience in the maternal and neonatal related study and this is addressed in the manuscript. 

2) What is defined as retained skills when a fraction of participants never had participated in training or theoretical background "25 percent had ever taken NR refresher training and 47.3% had ever performed NR. About half (50.1%) of study subjects ever worked in the delivery room". How was skill retention defined and quantified and what is meant by the expression of "good NR knowledge score" 

 Thank you, for the first two-sentence comment of #2, both midwives and nurses had basic theoretical and skill background during their higher education before graduation. Refreshers training is the additional rehearsal activity, not basic training. Ever performing NR and working in the delivery room were the expected factors that we tested for skill association. As we stated under the section of knowledge and skill measurement, we stated good skill retention when the participates scored the measurement tool 80% or higher. “good NR knowledge score” rephrased as good knowledge in NR.

3) What are the characteristics of the 4 health districts, any differences or similarities, in-hospital patients populations? Thank you for your clarification question, first, Eastern part of Ethiopia in the area with high fertility rate. Dire Dawa administration and Harari region are the independent regional state, mostly urban residents. However, Eastern and Western Harerghe zones are mostly from the rural Oromia regional states. In addition to these differences Dire Dawa administration is with the different nation, nationality, and different religion follower populations, which is directly governed by the federal government, Harari region is the smallest regional state in Ethiopia with a similar population with Dire Dawa administration, but not governed the regional state. The two West and East Hararghe zones have an almost similar population, which is most of them are Muslim in religion and single nation (Oromo). With these difference, the four health districts influence the capacity and readiness of the facility to give care to those different populations.

Their similarity is all the health districts follow the same health management system of the federal government and the health care providers from a similar multi-disciplinary for all health districts accordingly.

4) I would like to see the pre-tested, structured, observational checklist for evaluation of NR skills, as well as the adapted and modified structured questionnaire that was used and the scoring system. 

 Thank you for the inquiry. The copy of all the tools we used in this study is uploaded as supporting information. Since the participants’ working language is English, we have used only the English language version, and we didn’t have other language version. Since we extracted the tools from the different studies, which were validated in different studies by different body for both knowledge assessment and skill observation. Further, it is possible to find the studies used for the tools here below.

For knowledge assessment tools:

1. Kim Y, Ansari N, Kols A, Tappis H, Currie S, Zainullah P, et al. Assessing the capacity for newborn resuscitation and factors associated with providers’ knowledge and skills: a cross-sectional study in Afghanistan. BMC Pediatr [Internet]. 2013;13(1):140–52.

2. Averting Maternal Death and Disability Program (AMDD): Needs Assessment of Emergency Obstetric and Newborn Care (EmONC): Facilitator’s Guide. NewYork: Averting Maternal Death and Disability; 2009. 

3. Keyes, E.B., Haile‐Mariam, A., Belayneh, N.T., Gobezie, W.A., Pearson, L., Abdullah, M. and Kebede, H. (2011), Ethiopia's assessment of emergency obstetric and newborn care: Setting the gold standard for national facility‐based assessments. International Journal of Gynecology & Obstetrics, 115: 94-100. doi:10.1016/j.ijgo.2011.07.009

4. Abrha MW, Asresu TT, Araya AA, Woldearegay HG. Healthcare Professionals' Knowledge of Neonatal Resuscitation in Ethiopia: Analysis from 2016 National Emergency Obstetric and Newborn Care Survey. Int J Pediatr. 2019 Jul 16; 2019:8571351. https://doi.org/10.1155/2019/8571351

5. Mirkuzie AH, Sisay MM, Reta AT, Bedane MM. Current evidence on basic emergency obstetric and newborn care services in Addis Ababa, Ethiopia; a cross sectional study. BMC pregnancy and childbirth. 2014 Dec;14(1):354. doi:10.1186/1471-2393-14-354

For skill assessment tools:

1. Kim Y, Ansari N, Kols A, Tappis H, Currie S, Zainullah P, et al. Assessing the capacity for newborn resuscitation and factors associated with providers’ knowledge and skills: a cross-sectional study in Afghanistan. BMC Pediatr [Internet]. 2013;13(1):140–52.

2. Kinzie B, Gomez P. Basic maternal and newborn care. A guide for the skilled provider. 2004. Available at: http://www.healthynewbornnetwork.org/hnn-content/uploads/JHPIEGO_mnh.pdf.

3. Beck D, Ganges F, Goldman S, Long P. Saving Newborn Lives. Care of the Newborn Reference Manual. Washington, DC: Save the Children Federation. 2004;18. Available at: https://km.mohp.gov.np/sites/default/files/2018-07/Care_of_the_Newborn_Reference_Manual_2004.pdf

5) Further, a better description of how cluster sampling and simple random sampling methods were performed 

 Thank you. We used a cluster-sampling method first, based on their geographic location and administration into, Dire Dawa Administration (desert/hot climate and urban), Harari regional state (urban and middle climate region), East Hararghe and West Hararghe (rural part but, different in geographic location and administration). Then, we selected a total of 10 hospitals and 18 health centers from those clusters using a simple random sampling method from the list of public health facilities. Besides, a simple random sampling (lottery) method and proportional allocation to size were used to enroll midwives and nurses from the selected facilities. This description is incorporated into the manuscript.

6) Explain the differences in education and daily exposure to critically ill newborns, midwives vs nurses, 

 Thank you. Midwives and nurses are health care providers who can provide promotive, preventive, curative, and rehabilitative services at all levels of the health care system. Midwifery and nursing training in Ethiopia offered in both public and private per the ministry of science and higher education requirement: the diploma, post-basic, direct bachelor of sciences degree, MSc degree, and PhD level in both qualifications. Since nursing professionals are assigned to all units of the health facility, their education focuses on all medical care. Even if midwives takes all medical care education, lastly they focus on maternal and neonatal care. In addition to these, some midwives and nurses take in-service training that can help them to give quality care at where they assigned. Therefore, both midwives and nurses have access to managing critically ill newborns, midwives in labor and delivery ward and nurses in delivery, neonatology, and pediatric ward. 

7) 427 participated in the study. How many were invited and how were the invitation posted. Anyone who rejected participation 

.

 A total of 437 midwives and nurses were expected to be included in the assessment. Of these, 10 midwives and nurses were not included due to different reasons (four participants refused the invitation and the remaining six participants were inaccessible because they were in annual leave at the time of the assessment. Therefore, the assessment was limited to 427 (97.7%) of midwives and nurses those completely addressed by the assessment team. The assessment team visited each facility for one to three days to collect data if the study participants are unavailable. This description also addresses in the manuscript under study design and participants

8) Minors; Explain the NR unit. Fig 2 may be deleted as information appears in the text. Is there an IRB approval number?

 Thank you for your detailed evaluation. “Ever working in NR unit” is a typing error, now it updated to “ever performed NR”, which was correct on figure 1. This correction have dove throughout the document and Figure 2 is deleted. Our institution did not give IRB approval number since it is not online. The office notices the approval of the study by letter without approval number.

Thank you,

Authors

---

## [Editor Report · Decision Letter 1]

1 Jul 2020

Basic neonatal resuscitation skills of midwives and nurses in Eastern Ethiopia are not well retained: an observational study.

PONE-D-20-09955R1

Dear Dr. Yitagesu Sintayehu Abebe,

We’re pleased to inform you that your manuscript has been judged scientifically suitable for publication and will be formally accepted for publication once it meets all outstanding technical requirements.

Kind regards,

Georg M. Schmölzer

Academic Editor

PLOS ONE
---

## [Editor Report · Acceptance letter]

15 Jul 2020

PONE-D-20-09955R1 

Basic neonatal resuscitation skills of midwives and nurses in Eastern Ethiopia are not well retained: an observational study. 

Dear Dr. Abebe:

I'm pleased to inform you that your manuscript has been deemed suitable for publication in PLOS ONE. Congratulations! Your manuscript is now with our production department. 

Kind regards, 

on behalf of

Dr. Georg M. Schmölzer 

Academic Editor

PLOS ONE